# Evaluation of Dry Eye Disease Signs, Symptoms, and Vision-Related Quality of Life in Patients with Systemic Lupus Erythematosus

**DOI:** 10.3390/life15091423

**Published:** 2025-09-10

**Authors:** Wojciech Luboń, Anna Agaś-Lange, Ewa Mrukwa-Kominek, Adrian Smędowski, Dorota Wyględowska-Promieńska

**Affiliations:** 1Department of Ophthalmology, Faculty of Medical Sciences, Medical University of Silesia, 40-514 Katowice, Poland; 2Department of Ophthalmology, Professor K. Gibiński University Clinical Center of the Medical University of Silesia, 40-514 Katowice, Poland; 3Department of Pediatric Ophthalmology, Faculty of Medical Sciences, Medical University of Silesia, 40-514 Katowice, Poland; 4Department of Pediatric Ophthalmology, Professor K. Gibiński University Clinical Center of the Medical University of Silesia, 40-514 Katowice, Poland; 5GlaucoTech Co., 40-282 Katowice, Poland

**Keywords:** systemic lupus erythematosus, autoimmune, ocular complications, dry eye disease, quality of life

## Abstract

Dry eye disease (DED) represents one of the most prevalent ocular manifestations associated with systemic lupus erythematosus (SLE), with reported incidence rates ranging from 15% to 35%. DED constitutes a multifactorial condition that significantly impairs both visual function and health-related quality of life. The objective of this study was to assess the impact of DED symptoms on vision-related quality of life in patients diagnosed with SLE, employing the Ocular Surface Disease Index (OSDI) as a disease-specific instrument. Additionally, the study aimed to evaluate correlations between clinical diagnostic tests and OSDI scores, and to determine the frequency of abnormalities affecting individual ocular structures. This study included 35 SLE patients, identifying DED in 37.1%. Common ophthalmic abnormalities included lens opacification (22.9%) and hyaloid degeneration (34.3%). Astigmatism (>0.50 D cyl) was prevalent (60.0%), being significantly higher in DED patients. While visual acuity and intraocular pressure were comparable, DED patients showed significantly lower Schirmer I test values, reduced tear break-up time, and higher van Bijsterveld scores, indicating impaired tear film and surface integrity. OSDI scores were significantly elevated in the DED group, with 51.4% reporting moderate to severe dysfunction. Strong, statistically significant correlations between the OSDI and objective tear film parameters confirmed a robust association between subjective symptoms and clinical signs. These findings highlight the significant impact of DED on visual function in SLE patients, underscoring the importance of routine ophthalmological evaluation and timely intervention.

## 1. Introduction

Systemic lupus erythematosus (SLE) is a chronic, multisystemic autoimmune disorder characterized by widespread inflammation and tissue damage. It exerts a deleterious effect on connective tissue and affects numerous organ systems, including the visual system, leading to distinct ophthalmologic manifestations [1,2].

In addition, SLE presents with a broad spectrum of clinical manifestations due to immune dysregulation and multiorgan involvement. The disease commonly affects the skin, kidneys, joints, nervous system, and hematopoietic organs, while cardiovascular, pulmonary, gastrointestinal, and ocular complications are less frequent. The clinical course is highly variable, with periods of remission and exacerbation. Although the exact etiology remains unclear, SLE is thought to result from a complex interplay of genetic, environmental, and immunological factors. This systemic nature contributes to significant morbidity and a decline in health-related quality of life [3,4,5,6,7,8,9].

SLE occurs worldwide and exhibits a marked female predominance, with women accounting for 85–93% of cases. The median age of onset typically ranges from adolescence to early middle age. In females, disease incidence peaks between the third and fifth decades of life, whereas in males it occurs later, most commonly between the ages of 50 and 70. This sex-based difference in epidemiology is believed to be influenced by hormonal factors, particularly estrogen. Prevalence among women increases sharply at puberty and declines post menopause, while in men, the incidence increases gradually with age. The overall reported prevalence of SLE ranges from 20 to 150 cases per 100,000 individuals, with an annual incidence of 0.3–3.8 per 100,000 [10,11,12,13,14,15].

According to recent evidence, approximately 30% of SLE patients develop ophthalmic symptoms. These are frequently mild and self-limiting but may, in some instances, evolve into severe, sight-threatening complications. Ocular involvement typically reflects systemic disease activity and may precede or predict the involvement of other critical organ systems such as the cardiovascular, pulmonary, or renal systems [16,17].

Ocular manifestations may affect virtually any component of the eye, including the protective adnexa (eyelids), orbit, lacrimal apparatus, and intraocular structures such as the iris, ciliary body, vitreous body, retina, and optic nerve. The most reported condition is keratoconjunctivitis sicca (KCS), or dry eye disease (DED), affecting 25–35% of patients.

The primary objective of this study was to assess the impact of dry eye symptoms on vision-related quality of life in patients with systemic lupus erythematosus, using the validated Ocular Surface Disease Index (OSDI) questionnaire. Additional aims included evaluating correlations between OSDI scores and clinical ophthalmologic findings, as well as determining the prevalence of disorders affecting specific ocular structures in this patient population. Given the frequency of ocular involvement and its significant impact on daily functioning, this study provides a comprehensive evaluation of both subjective symptoms and objective ocular findings in SLE patients [18,19].

## 2. Materials and Methods

### 2.1. Study Design and Participants

Patients diagnosed and treated at the Department of Internal Diseases, Rheumatology and Clinical Immunology of the Upper Silesian Medical Center in Katowice, Poland, who had confirmed diagnoses of systemic lupus erythematosus (SLE) based on the American College of Rheumatology (ACR) criteria, underwent both basic and extended ophthalmological examinations. These examinations were conducted in the Cornea Unit of the Ophthalmology Department of the University Clinical Center, Prof. K. Gibiński in Katowice.

The study included 35 patients with a confirmed diagnosis of systemic lupus erythematosus (SLE), aged over 18 years, who underwent a complete ophthalmological assessment between June 2016 and December 2021. Inclusion criteria comprised confirmed SLE diagnosis established by a rheumatologist, an age of ≥ 18 years, and availability of complete clinical and ophthalmological data. Exclusion criteria were a history of ocular trauma or surgery, concurrent autoimmune or systemic diseases other than SLE, contact lens use, and incomplete ophthalmological records.

### 2.2. Ethical Approval

The study was approved by the Bioethical Committee of the Medical University of Silesia.

### 2.3. Ophthalmological Examination Procedures

Standard tests were performed in accordance with the tenets and guidelines of the Declaration of Helsinki. From June 2016 to December 2021, ophthalmological examinations were carried out in patients suffering from systemic lupus erythematosus. In total, 35 patients were included in the study, and a retrospective analysis of their examination results was performed. Each of the examined patients underwent a thorough ophthalmological analysis, which included the following tests, performed in the following specified order:Distant and near visual acuity test on Snellen charts with the best spectacle correction (BCDVA and BCNVA) in the same light conditionsEvaluation of the anterior and posterior segment of the eyeball in a slit biomicroscope (Haag-Streit, Switzerland) with a 90D lens ( Volk Optical Inc., Mentor, OH, USA)Examination of the retina using Optical Coherence Tomography (Zeiss, Germany)Evaluation of the eyeball using an ultrasonographic device (ultrasound of the eye with a 10 MHz probe).Evaluation of the refractive error with an RM-8100 autorefractometer (Topcon, Japan)Intraocular pressure (IOP) measurement with a Goldmann applanation tonometer (Haag-Streit, Switzerland)Evaluation of tear film stability using the Tear film Break-Up Time (T-BUT) test.Fluorescein staining test and anterior segment assessment using both the Oxford and Bijsterveld scaleEvaluation of tear secretion using the Schirmer test I without anesthesiaEvaluation of ocular surface dysfunction parameters using the Ocular Surface Disease Index (OSDI) questionnaire

All ophthalmological examinations were performed in a standardized sequence to avoid interference between tests, with a short resting interval of approximately 15 min maintained between the tear film break-up time test and the subsequent Schirmer I test.

### 2.4. Ocular Surface Disease Index (OSDI) Questionnaire

The Ocular Surface Disease Index (OSDI) questionnaire [20,21,22] was employed to quantitatively assess the impact of dry eye symptoms on vision-related quality of life in patients with systemic lupus erythematosus. This validated, disease-specific tool comprises 12 items subdivided into three domains: (1) ocular discomfort, capturing symptoms such as eye pain or irritation; (2) visual function, evaluating limitations in performing routine visual tasks, including reading, driving, or computer use; and (3) environmental triggers, assessing the exacerbating effects of external factors such as wind, air conditioning, or smoke on dry eye symptoms.

Participants were instructed to respond based on a 7-day recall period, rating the frequency of symptoms using a 5-point Likert scale: “none of the time”, “some of the time”, “half of the time”, “most of the time”, or “all of the time.”

The overall OSDI score was computed using the standardized formula:$$\mathrm{OSDI}\;\mathrm{score}\;=\;\frac{(sum\;of\;scores\;for\;all\;answered\;questions)\times 100}{(total\;number\;of\;questions\;answered)\times 4}$$

Subscale scores for each domain were derived using the same formula, applied separately to the specific questions related to that subscale. Thus, both the overall and domain-specific scores ranged from 0 to 100, where higher values indicated more severe symptoms and greater functional impairment.

Scoring interpretation was performed in accordance with established guidelines, classifying symptom severity as follows:0–12: Normal13–22: Mild DED23–32: Moderate DED33–100: Severe DED

This instrument was selected for its widespread clinical use, high sensitivity in detecting dry eye symptoms, and robust psychometric properties. The OSDI score reflects how often patients experience dry eye symptoms in daily life and how much these symptoms interfere with vision-dependent tasks. Higher scores indicate more frequent symptoms and greater functional limitations.

### 2.5. Tear Film Break-Up Time (T-BUT) Test

The tear film break-up time (T-BUT) test was performed to assess the stability of the pre-corneal tear film, serving as an indicator of ocular surface health and tear film integrity. A single drop of 2% sodium fluorescein solution was instilled into the lower conjunctival fornix of each patient. Using slit-lamp biomicroscopy equipped with a cobalt blue filter, the time interval between a complete blink and the appearance of the first dry spot or discontinuity in the tear film was recorded.

This moment is visualized as dark lines or spots appearing on the otherwise uniform greenish-yellow fluorescein background (Figure 1). The test was repeated three times per eye, and the mean value was used for analysis to minimize intra-examiner variability.

The following interpretation criteria were applied in accordance with current ophthalmologic standards [23]:T-BUT of >10 s was considered normal tear film stability;T-BUT of <10 s indicated decreased tear film stability, often associated with lipid layer dysfunction;T-BUT of <5 s was interpreted as indicative of significant ocular surface lubrication deficiency and marked tear film instability, typically resulting from meibomian gland dysfunction or lipid layer insufficiency.

This test is non-invasive, reproducible, and widely used in both clinical and research settings to diagnose dry eye disease (DED) and assess treatment efficacy. This test evaluates how long the tear film remains stable after a blink. A shorter time means the tear film breaks up too quickly, leading to blurred vision, discomfort, and an unstable ocular surface, which are typical features of dry eye disease.

### 2.6. Schirmer I Test

A conventional Schirmer I test without topical anesthesia was performed to evaluate the basal and reflex secretion of the aqueous component of the tear film. The test involved the placement of a standardized filter paper strip (Whatman no. 41) folded at the notch and inserted into the lateral third of the lower conjunctival fornix. The patients were instructed to gently close their eyelids during the 5 min test duration.

After the elapsed time, the extent of wetting on the strip was measured in millimeters from the fold. This value reflects the secretory function of the lacrimal glands.

Interpretation of results was based on established clinical criteria [24]:Wetting of >15 mm: Normal aqueous tear secretionWetting of 10–15 mm: Early or borderline aqueous deficiencyWetting of 5–10 mm: Moderate aqueous tear deficiency, suggestive of evolving dry eyeWetting of <5 mm: Severe aqueous-deficient dry eye, indicating advanced lacrimal hyposecretion and significant tear film insufficiency

This test provides a quantitative assessment of tear production and is particularly useful in diagnosing aqueous-deficient dry eye subtypes, such as those associated with Sjögren’s syndrome or systemic autoimmune diseases like SLE.

### 2.7. Ocular Surface Staining Assessment

Assessment of ocular surface damage was conducted using vital dye staining techniques, specifically rose Bengal and fluorescein dyes, which are widely accepted in the evaluation of dry eye disease (DED).

To assess conjunctival damage, a rose Bengal-impregnated strip was gently applied to the inferotemporal bulbar conjunctiva. The resulting staining pattern was evaluated using the van Bijsterveld scoring system, which assesses three ocular surface zones: the nasal and temporal conjunctiva and the cornea. Each zone is scored from 0 to 3, with a maximum possible score of 9. Higher scores indicate more extensive epithelial damage and greater severity of ocular surface disease.

Corneal epithelial damage was further graded using the Oxford grading scheme, a standardized tool for quantifying fluorescein staining of the corneal and conjunctival epithelium. This semi-quantitative scale classifies staining into six categories, ranging from grade 0 (no staining) to grade 5 (severe punctate epithelial damage). Evaluation was based on a reference chart comprising panels A through E, where staining is illustrated with colored dots. The intensity and extent of staining in patient eyes were compared directly with the reference images.

In this system, each panel represents a progressive increase in staining intensity: staining increases by approximately one logarithmic unit between panels A and B and by 0.5 logarithmic units between panels B through E.

Following dye instillation, ocular surface staining was examined under a slit-lamp biomicroscope (Haag-Streit) at standardized settings (×16 magnification) and with appropriate light filters to optimize visualization. It should be noted that the dye highlights areas of epithelial damage on the ocular surface. The more staining that appears, the more severe the ocular surface impairment. Bright fluorescent spots on the cornea correspond to areas where the protective tear film is insufficient.

### 2.8. Diagnostic Criteria for Dry Eye Disease

In this study, abnormal values in ocular surface tests were defined as follows:Schirmer I test: ≤5 mm of wetting after 5 minTear Break-Up Time (T-BUT): <10 svan Bijsterveld score: >3

A diagnosis of dry eye disease (DED) was established when at least two of the above parameters met these defined abnormality criteria, in accordance with previously validated diagnostic protocols in autoimmune populations [25,26].

This multimodal assessment approach enabled a comprehensive evaluation of tear quantity, tear film stability, and epithelial integrity, providing reliable criteria for the diagnosis of DED in systemic lupus erythematosus patients.

### 2.9. Statistical Analysis

All statistical analyses were performed using validated statistical software (Statistica, Version 13.3 TIBCO Software Inc., Palo Alto, CA, USA). A significance level of α = 0.05 was adopted for all statistical tests. Where appropriate, adjustments for multiple comparisons were applied using the Bonferroni correction or its non-parametric equivalents. The distribution of continuous variables was assessed using the Shapiro–Wilk test to determine the assumption of normality. Homogeneity of variances was evaluated with Levene’s test.

In cases where the data met the assumptions of parametric testing, statistical comparisons were performed using one-way analysis of variance (ANOVA), followed by Bonferroni post hoc testing when applicable. For comparisons between two groups, the independent samples Student’s *t*-test was applied.

When assumptions for parametric analysis were not met, non-parametric alternatives were used: the Kruskal–Wallis test with Dunn’s post hoc correction for multiple group comparisons, and the Mann–Whitney U test for two-group comparisons.

The strength and direction of linear associations between continuous variables were evaluated using Pearson’s correlation coefficient (r) for normally distributed data or Spearman’s rank correlation coefficient (rho) for non-normally distributed variables.

Categorical variables were analyzed using the chi-squared (χ^2^) test of independence. In cases where expected cell counts were below 5, Fisher’s exact test was used instead.

Results were presented with corresponding *p*-values and confidence intervals where applicable. In accordance with commonly accepted practice in ophthalmological research, when both eyes were examined, the mean value of the parameters obtained from the right and left eye was calculated for each patient, in order to minimize the potential influence of interocular asymmetry and to avoid overrepresentation of individual subjects in the analysis.

## 3. Results

### 3.1. Characteristics and Clinical Findings

The study group consisted of 35 patients (70 eyes) diagnosed with systemic lupus erythematosus (SLE), including 33 women (94.3%) and 2 men (5.7%). The mean age was 46 years, with a range from 24 to 65 years (M = 46, SD = 10.75).

Based on the diagnostic criteria established in this study, dry eye disease (DED) was confirmed in 13 patients (37.1%), while 22 patients (62.9%) did not meet the criteria for DED, constituting the non-DED subgroup.

Additional ophthalmological abnormalities identified in the cohort included

Lens opacification in 8 patients (22.9%);Hyaloid degeneration in 12 patients (34.3%);Retinal degenerations, including dry (atrophic) changes in 4 patients (11.4%) and wet (exudative) changes in 1 patient (2.9%);Hypertensive or microangiopathic retinal angiopathy in 8 patients (22.9%);Episcleritis in 1 patient (2.9%).

No cases of retinitis or retinal vasculitis were observed in the examined group. No statistically significant association was found between the presence of posterior segment abnormalities and the occurrence of dry eye disease.

These findings are summarized in Table 1.

Among the 35 patients, refractive errors were common, with astigmatism of >0.50 diopters (D cyl) identified in 21 patients (60.0%). Notably, patients diagnosed with DED demonstrated significantly higher degrees of astigmatism.

Furthermore, 22 patients (62.8%) reported subjective symptoms suggestive of ocular surface dysfunction, including sensations of itching, burning, foreign body presence, dryness, and ocular redness.

These symptoms were more prevalent in the DED group, supporting the correlation between clinical signs and patient-reported symptoms.

Table 2 presents a comparative analysis of the study population, stratified into two groups: patients diagnosed with dry eye disease (DED) and those without DED, based on clinical examination and OSDI scores—both total and subscale-specific.

There were no statistically significant differences between the groups regarding demographic variables. The mean age did not differ significantly between DED and non-DED patients (*p* = 0.111), and no association with sex distribution was observed (*p* = 0.600).

The best-corrected distance visual acuity (BCDVA) measured using the Snellen decimal scale was on average 0.896 and 0.905 for the right and left eye, respectively, for all patients studied (*n* = 35). In isolated cases (4.3%), the BCDVA was less than 0.5. There were no statistically significant differences in the BCDVA between individual groups of patients (*p* = 0.700).

The median of the best-corrected near visual acuity (BCNVA) was in Snellen scale 0.50 (SD = 0.26) for both the right and left eye in all examined patients (*n* = 35). No statistically significant differences were noted in the BCNVA between individual patient groups (*p* = 0.76).

The mean intraocular pressure (IOP) was 16.4 mmHg (right eye) and 16.1 mmHg (left eye), with no statistically significant differences found between the DED and non-DED groups (*p* = 0.520).

### 3.2. Tear Film Parameters and Quality of Life

In contrast, statistically significant differences were observed in the diagnostic tests assessing the quantity and stability of the tear film.

Patients with DED exhibited significantly lower values in the Schirmer I test (*p* = 0.0003) andSignificantly reduced tear break-up time (T-BUT) (*p* = 0.0002) compared with non-DED counterparts.

These findings confirm impaired tear production and instability of the tear film in DED patients, reflecting greater anterior segment pathology.

In addition, ocular surface staining assessments revealed notable abnormalities in the DED group:van Bijsterveld scores were significantly higher among DED patients (*p* = 0.0000);The Oxford grading scale demonstrated a trend toward statistical significance (*p* = 0.080), indicating a possible association.

These results underscore the relevance of ocular surface staining techniques in identifying subclinical inflammation and epithelial damage in SLE patients with DED.

Based on these findings, topical therapeutic intervention with lubricants and/or anti-inflammatory agents appears warranted in most DED cases, aiming to restore ocular surface integrity and improve quality of life.

The following mean test values were recorded across the entire cohort:Schirmer I Test: 10.5 mm (SD = 4.17)Tear Break-Up Time (T-BUT): 8.5 s (range: 6.5–11.5 s)van Bijsterveld score: Median of 3 (range: 1–4)

The Ocular Surface Disease Index (OSDI) total and subscale scores are summarized in Table 2 and Figure 2. Statistically significant differences were observed between the DED and non-DED groups for the overall OSDI score as well as each subscale, indicating a greater subjective burden of disease among DED patients.

Specifically, 18 patients (51.4%) scored >22 points in the OSDI questionnaire, suggesting moderate to severe ocular surface dysfunction. Most of these patients (61.1%) belonged to the DED group. Detailed group-wise OSDI results are shown in Table 3.

Correlation analyses between the OSDI (total and subscale scores) and objective clinical parameters—T-BUT, Schirmer I, and ocular surface staining scores (van Bijsterveld and Oxford)—are presented in Table 4.

Statistically significant and clinically meaningful correlations were observed in nearly all comparisons, confirming strong associations between subjective symptom severity and objective diagnostic test results.

In this study, statistically significant correlations were identified between the total OSDI score and the results of the T-BUT test, Schirmer I test, and ocular surface staining scores using both the van Bijsterveld and Oxford grading systems.

The relationships between ocular surface dysfunction parameters and tear secretion measures with the OSDI total and subscale scores were predominantly moderate in strength and reached statistical significance in most instances. These correlations confirm the clinical relevance of combining subjective symptom evaluation with objective diagnostic testing.

The results collectively underscore the negative impact of ocular surface abnormalities on visual function and the daily activities of patients with systemic lupus erythematosus. Specifically, patients with poorer tear film stability and reduced aqueous secretion reported significantly greater limitations in activities involving reading, screen use, and exposure to environmental triggers.

According to the statistical analysis, ocular surface damage had the most pronounced effect on the OSDI subscale related to visual symptoms, reflecting discomfort and degradation in the quality of vision during everyday tasks (see Figure 3).

These findings highlight the multidimensional burden of dry eye disease in SLE and provide evidence for the importance of routine ophthalmological evaluation and early therapeutic intervention in this patient population.

## 4. Discussion

The present study demonstrated that patients with systemic lupus erythematosus (SLE) frequently exhibited abnormalities of tear film stability and secretion, as evidenced by reduced T-BUT and Schirmer I values, as well as increased ocular surface staining. In addition, the OSDI scores indicated a significant impact of dry eye symptoms on vision-related quality of life in this population. Importantly, correlations between the OSDI results and objective clinical tests confirmed that subjective symptoms were strongly associated with measurable ocular surface dysfunction. These findings underline the relevance of ocular involvement in SLE and emphasize the need for targeted ophthalmological evaluation in this patient group.

Dry eye syndrome, also known as dry eye disease (DED) or keratoconjunctivitis sicca (KCS), is one of the most frequent causes of ophthalmologic consultations in the general population. According to the updated Tear Film and Ocular Surface Society Dry Eye Workshop III (TFOS DEWS III) report, DED is defined as the following:

*“A disorder of the ocular surface characterized by tear film instability and hyperosmolarity, ocular surface inflammation and damage, and neurosensory abnormalities, leading to symptoms of discomfort, visual disturbance, and reduced quality of life.”* [27]

This definition underscores the complex pathophysiology of DED, involving both structural and neuroimmunological components.

The prevalence of DED in this study was 37.1%, aligning with previously reported rates ranging from 15% to 35% [28,29,30,31]. In our cohort, 22 out of 35 patients (62.8%) reported dry eye symptoms, most commonly itching, burning, foreign body sensation, dryness, and redness of the conjunctiva. This is consistent with the literature, where 18–60% of SLE patients present with similar ocular complaints [32].

The Ocular Surface Disease Index (OSDI) is a well-validated instrument designed to evaluate the frequency and severity of dry eye symptoms and their impact on daily activities and work productivity. Although systemic disease may affect overall quality of life, the OSDI remains relatively specific to ocular surface symptoms. It has proven valuable in discriminating between healthy individuals and those with varying severities of DED and is widely used in both clinical trials and epidemiological research [33]. In this study, higher OSDI scores were clearly associated with objective indicators of ocular surface damage, reinforcing its utility in SLE-related DED assessment.

In the study by Jensen et al., 60% of SLE patients (*n* = 20) reported at least one symptom of DED, including mild irritation, foreign body sensation, grittiness, conjunctival redness, and dryness [34]. These symptoms are consistent with the subjective complaints recorded in our study. Furthermore, a subset of patients with SLE and DED developed secondary Sjögren’s syndrome (sSS). Manoussakis et al. identified sSS in 9.2% of a cohort of 283 SLE patients, highlighting the need to consider overlapping autoimmune syndromes when assessing ocular surface pathology [28].

It has been shown in the medical literature that patients with SLE have an increased incidence and rate of developing acute conjunctivitis, which is a disease most often caused by a bacterial or viral infection of the conjunctiva. It also seems to be related to DED due to the generalized impairment of the body’s immune responses in patients with SLE [29].

The pathogenesis of KCS in SLE is believed to involve fibrosis of the conjunctiva and lacrimal glands, leading to aqueous tear deficiency and secondary ocular surface damage [35,36]. Elevated levels of pro-inflammatory mediators, including IL-17, have been identified in the tear film of affected individuals. These are the same markers present in autoimmune diseases that cause inflammation processes leading to scarring such as Steven Johnson’s syndrome [37,38].

Clinical signs of ocular surface damage in SLE may include symblepharon formation and exposure keratopathy. Histopathological examination of conjunctival biopsy specimens often reveals goblet cell loss, epithelial keratinization, mononuclear cell infiltration, and granuloma formation. Immunopathologic findings include immune complex deposition along the epithelial basement membrane, along with increased infiltration by CD4^+^ and CD8^+^ T lymphocytes, B lymphocytes, and macrophages [39,40]. Although DED rarely leads to complete visual loss, its chronic course and recurrent symptoms significantly impair visual function and reduce patient well-being in daily and social activities [41,42].

Moreover, DED exerts a profound impact on multiple dimensions of patients’ quality of life (QoL), including physical, social, emotional, and occupational domains [43,44]. Emerging evidence suggests that DED may be conceptualized as a chronic pain syndrome, wherein persistent ocular discomfort contributes to psychological burden and functional disability [42,44,45].

A notable limitation of this study is that the analyzed patient population consisted of individuals receiving chronic immunosuppressive therapy for systemic lupus erythematosus, including glucocorticosteroids and antimalarials such as hydroxychloroquine (HCQ). Yavuz et al. have shown that HCQ may contribute to ocular surface damage in patients with autoimmune disorders [46]. Thus, the potential iatrogenic effects of long-term systemic therapies must be considered when interpreting the prevalence and severity of DED in this population.

In a study by Wang et al. (2019), parameters such as T-BUT, tear meniscus height, and the OSDI score were found to correlate significantly, albeit weakly, with SLE activity [32]. Similarly, Bartlett et al. reported that the strength of correlations between DED diagnostic tests and subjective symptoms typically ranged from –0.4 to 0.4, indicating low to moderate concordance [33]. However, in the present study, we observed stronger correlations between subjective (OSDI) and objective (Schirmer I, T-BUT, Bijsterveld, and Oxford) indicators, suggesting a more direct relationship between clinical signs and symptomatology in this SLE population.

The diagnostic approach in this study was based on three primary clinical tools: T-BUT, Schirmer I, and the van Bijsterveld scoring system—all widely accepted in the ophthalmologic assessment of DED [25,47]. Nonetheless, there is growing recognition that additional diagnostic methods, such as tear film osmolarity testing, may offer enhanced sensitivity and specificity. Jacobi et al. demonstrated that osmolarity measurement is among the most sensitive markers of tear film dysfunction [48]. Furthermore, in the present study, T-BUT was assessed using fluorescein under slit-lamp biomicroscopy, as non-invasive ocular surface analyzers were not available in our clinical setting during the study period. Although fluorescein T-BUT remains a standard and validated test, non-invasive techniques may provide greater repeatability and reduce observer-dependent variability. Future research should therefore incorporate both osmolarity assessment and non-invasive tear film stability analysis to refine DED diagnostics in autoimmune populations. In addition, the study did not include an assessment of meibomian gland function, which represents an important factor in the pathophysiology of evaporative dry eye. Future studies should incorporate a standardized evaluation of meibomian gland morphology and function to provide a more comprehensive understanding of ocular surface involvement in SLE patients.

Beyond these established methods, the early detection of DED in high-risk groups such as SLE patients may benefit from incorporating non-invasive diagnostic modalities. Techniques such as Non-Invasive tear Break-Up Time (NI-BUT), tear film osmolarity measurement, and meibography for the assessment of meibomian gland structure and function have been shown to improve diagnostic sensitivity and repeatability compared with conventional tests. Integrating these tools into both research protocols and clinical practice may enable earlier recognition of ocular surface impairment, allowing timely initiation of treatment before irreversible damage develops.

Of note, the study cohort consisted predominantly of female patients, many of whom were premenopausal or menopausal, representing a group at higher baseline risk for dry eye disease. This gender imbalance should be taken into account when interpreting the findings, as it may limit their generalizability to the broader SLE population. Although age is a well-known risk factor for dry eye disease, no statistically significant differences in age were found between patients with and without DED in this study. This may suggest that disease duration alone is not the primary determinant of ocular surface involvement in SLE. Nevertheless, the absence of complete data on SLE duration precluded a definitive analysis of its association with dry eye disease, and this represents a limitation of the present study.

Another important limitation of the present study is the relatively small number of SLE patients included, which reduces the statistical power and external validity of the findings. Furthermore, the lack of an age- and sex-matched control group from the general population prevents us from determining whether the prevalence and severity of dry eye disease observed here are specific to systemic lupus erythematosus or may overlap with background rates in otherwise healthy individuals. Future research including larger cohorts and appropriate controls is required to clarify these aspects and strengthen the external validity of the results.

Given the close association between ocular surface impairment and functional limitations, the primary therapeutic goal in managing DED should be the improvement of patient quality of life, beyond the mere correction of clinical signs. This patient-centered approach underscores the broader significance of DED—both individually and societally.

There is a clear need to enhance public awareness regarding the prevalence and consequences of dry eye, particularly in individuals with chronic autoimmune conditions. Educational campaigns and interdisciplinary care models may facilitate earlier recognition and timely intervention.

Ophthalmologists and other healthcare providers should maintain a high index of suspicion for DED in patients with systemic autoimmune diseases such as SLE and proactively assess its impact on visual function and daily living, integrating both subjective and objective evaluation tools in clinical practice.

## 5. Conclusions

Dry eye disease (DED) is a prevalent and increasingly recognized condition that leads to a multifactorial decline in visual performance and quality of life (QoL). This study summarizes the burden of DED in patients with systemic lupus erythematosus (SLE), with a specific focus on its functional impact on daily visual tasks and patient-reported outcomes.

The findings are consistent with prior research, demonstrating that DED significantly affects multiple dimensions of QoL, particularly the ability to engage in visually demanding activities, such as reading and driving, as well as workplace productivity, especially when screen exposure is prolonged. These effects may translate into broader socioeconomic consequences, emphasizing the public health relevance of recognizing and managing DED effectively.

In our study, patients with SLE and DED showed statistically significantly lower vision-related quality of life, as reflected by elevated scores in the Ocular Surface Disease Index (OSDI) questionnaire, compared with those without DED. Furthermore, the OSDI scores observed in this cohort aligned with those reported in previous studies of SLE patients with moderate DED and were lower than those reported for patients with more severe forms of ocular surface disease, such as advanced Sjögren’s syndrome (both primary and secondary) or systemic sclerosis [25,49].

Our findings underscore the critical need for proactive screening and comprehensive management of DED in SLE patients to mitigate its substantial impact on their daily lives and overall well-being.

## Figures and Tables

**Figure 1 life-15-01423-f001:**
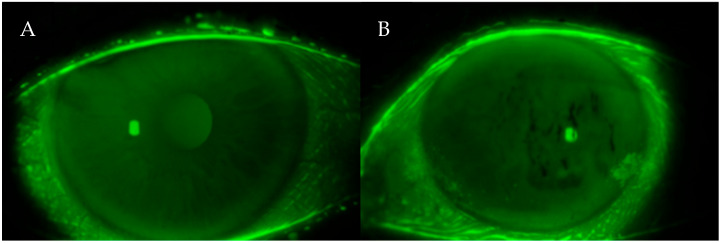
(**A**,**B**) Fluorescein photographs of the ocular surface during the T-BUT test. The dark spots visible on the corneal surface indicate areas where the tear film has broken up after blinking. These irregularities represent tear film instability, a hallmark of dry eye disease.

**Figure 2 life-15-01423-f002:**
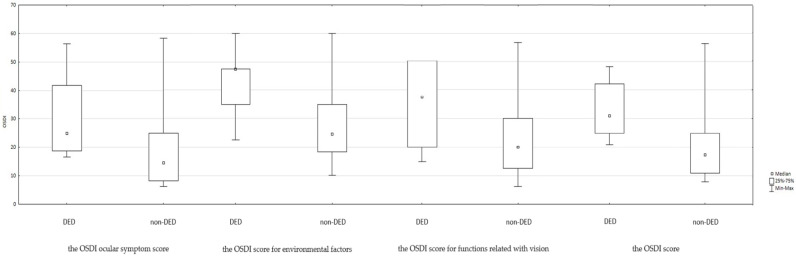
Summary of the OSDI score in DED groups.

**Figure 3 life-15-01423-f003:**
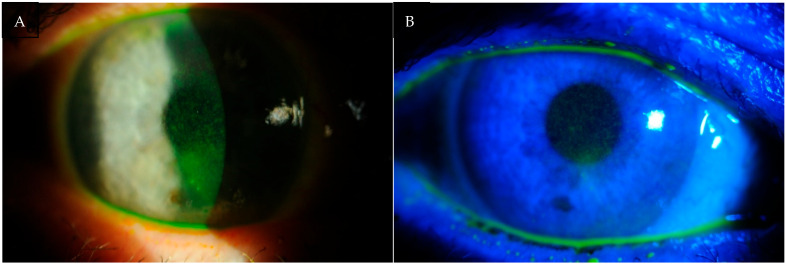
(**A**,**B**) Photographs of the anterior segment of the eye of patients with DED. Fluorescein staining highlights epithelial damage of the ocular surface, visible as bright punctate spots. The greater the staining intensity and extent, the more severe the ocular surface disease.

**Table 1 life-15-01423-t001:** Ophthalmic abnormalities among patients—percentage distribution.

Ophthalmic Abnormalities	Number of Patients (%)
Lens opacification	8 (22.9%)
Cataract	1 (2.9%)
Degenerative changes in the vitreous body	12 (34.3%)
Retinoschisis/retinal detachment	0
Degenerative “dry” changes in the retina	4 (11.4%)
Exudative or hemorrhagic changes in the retina	1 (2.9%)
Abnormalities in the course and size of blood vessels (narrowing or widening of veins and arteries)	8 (22.9%)
Visual field defects	5 (14.3%)
Episcleritis	1 (2.9%)
Refractive error: astigmatism (>0.50 D cyl)	21 (60.0%)

**Table 2 life-15-01423-t002:** Comparison of ophthalmic parameters in the DED and non-DED groups. The measures of location and dispersion appropriate for the tests used are given as the following: a—Student’s *t*-test, mean (standard deviation); b—Mann–Whitney U test, median (quartile range); asterisk—statistically significant result.

	DED (n = 13)	Non-DED (n = 22)	*p*	Total (n = 35)
Schirmer Test [mm/5 min]	7.4 (3.11)	12.3 (3.64)	0.0003 ^a,^*	10.5 (4.17)
Oxford scale	2.4 (0.79)	1.7 (1.27)	0.0818 ^a^	1.9 (1.16)
T-BUT [s]	6 (5.5–7.5)	11.3 (7–12)	0.0002 ^b,^*	8.5 (6.5–11.5)
Bijsterveld staining	4 (4–4)	2 (1.0–2)	0.0000 ^b,^*	3 (1–4)
OSDI	31 (25–42)	17.5 (11–25)	0.0007 ^b,^*	25 (14–31)
OSDI vision-related function	37 (20–50)	20 (12.5–30)	0.0123 ^b,^*	25 (15–40)
OSDI ocular symptoms	25 (18.8–41.7)	14.6 (8.3–25)	0.0036 ^b,^*	18.5 (12.5–25)
OSDI environmental triggers	37.5 (25–37.5)	14.6 (8.3–25)	0.0049 ^b,^*	25 (12.5–37.5)

**Table 3 life-15-01423-t003:** Summary of OSDI stage count and frequency with Fisher’s exact test (two-tailed).

OSDI	Non-DED	DED	Sum—Rows	Fisher’s Exact Test
Normal	6 27%	0 0%	6 17%	*p* = 0.0645
Mild	9 41%	2 15%	11 31%	*p* = 0.1501
Moderate	5 23%	5 38%	10 29%	*p* = 0.4437
Severe	2 9%	6 46%	8 23%	*p* = 0.3209
Sum—Columns	22	13	35	

**Table 4 life-15-01423-t004:** Spearman ρ correlation of OSDI variables. Asterisks indicate statistically significant results.

	Schirmer Test		T-BUT		Bijsterveld		Oxford Scale	
	**ρ**	* **p** *	**ρ**	* **p** *	**ρ**	* **p** *	**ρ**	* **p** *
OSDI	−0.4286	0.0102 *	−0.7203	0.0000 *	0.6072	0.0001 *	0.7844	0.0000 *
Vision-related function	−0.2925	0.0882	−0.5939	0.0002 *	0.4069	0.0153	0.6958	0.0000 *
Ocular symptoms	−0.4596	0.0055 *	−0.6365	0.0000 *	0.5207	0.0013 *	0.6068	0.0001 *
Environmental triggers	−0.4021	0.0167	−0.5798	0.0003 *	0.5998	0.0001 *	0.7262	0.0000 *

## Data Availability

The data used to support the findings of this study are available from the corresponding author upon request.

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
