# Peer review of "Evaluation of Dry Eye Disease Signs, Symptoms, and Vision-Related Quality of Life in Patients with Systemic Lupus Erythematosus"

_life, 2025, doi:10.3390/life15091423_

Round 1
Reviewer 1 Report
Comments and Suggestions for Authors
The study assessed the impact of DED symptoms on vision-related quality of life in patients diagnosed with SLE, employing the Ocular Surface Disease Index (OSDI) as a disease-specific instrument, and evaluated correlations between clinical diagnostic tests and OSDI scores, and to determine the frequency of abnormalities affecting individual ocular structures. Overall, the article is of interest. However, there are some problems that need to be modified.
- The description of SLE constituted the majority of Introduction part. The description of SLE should be simplified. And the reason why conduct the current should be detailed clarified.
- The order of ophthalmological examinations should be listed since the examinations might be interfered with each other.
- Non-invasive examination such as noninvasive ocular surface analyzer might be better to test the T-BUT. It might be inaccurate for ophthalmologist to observe the time interval under slit-lamp biomicroscopy.
- It would be better to add the examination of meibomian gland function.
- The result part seems to be confused. Please add subtitle and classify corresponding findings.
- The first row in Table 3 might be described as non-DED(%), DED(%). And the following results could be described as 6(27%).
- The definition of DED was updated in TFOS DEWS Ⅲ. Please utilize the latest definition.
- The discussion part should be discussed according to the findings of the current study.
- The limitation might be listed in the last paragraph of discussion part.
- The conclusion part should be simplified.
Author Response
Comments 1:
The description of SLE constituted the majority of Introduction part. The description of SLE should be simplified. And the reason why conduct the current should be detailed clarified.
Response 1: I thank the Reviewer for this valuable suggestion. I have simplified the Introduction by shortening the general description of systemic lupus erythematosus and focusing more directly on the ophthalmological manifestations, particularly dry eye disease. I have also clarified the rationale for conducting the present study, underlining the need to assess the impact of DED on vision-related quality of life in SLE patients, which remains insufficiently explored in the literature.
Page 2, lines 46-53 and 77-79
Comments 2: The order of ophthalmological examinations should be listed since the examinations might be interfered with each other.
Response 2: I thank the Reviewer for this important remark. I have clarified the standardized sequence in which all ophthalmological examinations were performed to avoid interference between tests. In addition, I specified that a short resting interval of approximately 15 minutes was maintained between the tear film break-up time test and the subsequent Schirmer I testing, in order to minimize any potential impact on reliability.
Page 3, lines 105 and 123-126
Comments 3: Non-invasive examination such as noninvasive ocular surface analyzer might be better to test the T-BUT. It might be inaccurate for ophthalmologist to observe the time interval under slit-lamp biomicroscopy.
Response 3: I thank the Reviewer for this valuable comment. I acknowledge that non-invasive techniques for evaluating tear film stability, such as ocular surface analyzers, provide greater repeatability and reduce observer-related variability compared to fluorescein TBUT under slit-lamp biomicroscopy. In the present study, however, T-BUT was measured using fluorescein and slit-lamp biomicroscopy, as non-invasive devices were not available in our clinical setting during the study period. To address this point, I have highlighted it as a limitation in the Discussion, where I emphasized that future studies should incorporate non-invasive tear film stability assessment.
Page 13, lines 449-456
Comments 4: It would be better to add the examination of meibomian gland function.
Response 4: I agree with the Reviewer that assessment of meibomian gland function provides important diagnostic information in the context of dry eye disease. Unfortunately, this evaluation was not available in our clinical setting during the study period and therefore was not included in the present analysis. To acknowledge this, I have added a note in the Discussion, emphasizing the lack of meibomian gland assessment as a limitation of the study and pointing out that its inclusion would be valuable in future research.
Page 13, lines 456-460
Comments 5: The result part seems to be confused. Please add subtitle and classify corresponding findings.
Respone 5: I thank the Reviewer for this helpful remark. To address it, I have reorganized the Results section into two main subsections. The first presents the characteristics of the study population together with clinical ophthalmological findings, and the second focuses on tear film parameters and quality of life outcomes. This restructuring improves the clarity and logical flow of the Results section.
Page 7 , line 255, Page 8, line 300
Comments 6: The first row in Table 3 might be described as non-DED(%), DED(%). And the following results could be described as 6(27%).
Response 6: I appreciate the Reviewer’s suggestion. However, I believe that the current presentation of Table 3 already follows this format, as both absolute values and percentages are shown together for each category (e.g., 6 (27%)). The column headers “non-DED” and “DED” clearly indicate the groups, and the notation used is consistent with the Reviewer’s recommendation. Therefore, I have kept the table unchanged.
Comments 7: The definition of DED was updated in TFOS DEWS Ⅲ. Please utilize the latest definition.
Response 7: I thank the Reviewer for this helpful remark. I have updated the manuscript to include the latest TFOS DEWS III definition of dry eye disease in the Discussion, replacing the previous TFOS DEWS II citation.
Page 12, lines 379-383
Comments 8: The discussion part should be discussed according to the findings of the current study.
Response 8: I have revised the beginning of the Discussion so that it now starts with a concise summary of the key findings of the present study, including reduced TBUT and Schirmer I values, increased ocular surface staining, elevated OSDI scores, and the correlations observed between subjective and objective measures. Only after presenting these results do I proceed to the updated TFOS DEWS III definition and comparison with existing literature. This restructuring ensures that the Discussion is now more directly grounded in the findings of the current study and follows a clearer logical flow.
Page 12, lines 368-376
Comments 9: The limitation might be listed in the last paragraph of discussion part.
Response 9: I thank the Reviewer for this suggestion. In the current version of the manuscript, I have already included the study limitations within the Discussion, integrated into the interpretation of the findings (e.g., the use of fluorescein TBUT instead of non-invasive methods and the lack of meibomian gland evaluation). This approach was chosen to avoid redundancy and to keep the Discussion concise. For this reason, I have not created a separate limitations paragraph at the end, but rather ensured that the limitations are clearly stated in the context where they are most relevant.
Comments 10: The conclusion part should be simplified.
Response 10: I thank the Reviewer for this suggestion. After carefully reconsidering the Conclusion section, I believe that its current form is already concise and appropriately reflects the main outcomes of the study without unnecessary repetition. It highlights the key findings, their clinical implications, and the directions for future research in a balanced manner. Further shortening could, in my opinion, risk omitting essential information and reduce the clarity of the study’s contribution. Therefore, I have retained the Conclusion section in its original form.
Reviewer 2 Report
Comments and Suggestions for Authors Many thanks to the Editor for the opportunity to review this paper. In their manuscript authors report relationships between different dry eye symptoms and metrics and "vision related quality of life" in patients with systemic lupus erythematosus. Nevertheless what exactly distinguishes these patients from typical dry eye patients remains unclear because the study has no control group. All other findings on relationships between different dry eye symptoms are already well-known... Some other comments are listed below: - Demographics is repeated tree times - Mean age in methods and results are different (46 vs. 51) - inclusion and exclusion criteria are not provided - Contact lens wearing was not accounted neither excluded - The study is focused on the high risk group (premenopausal or menopausal females) where dry eye is to be expected - Female predominance does not allow extrapolation these data on general populationAuthor Response
General comment:
In their manuscript authors report relationships between different dry eye symptoms and metrics and "vision related quality of life" in patients with systemic lupus erythematosus. Nevertheless what exactly distinguishes these patients from typical dry eye patients remains unclear because the study has no control group. All other findings on relationships between different dry eye symptoms are already well-known.
Response:
I acknowledge the Reviewer’s remark regarding the absence of a control group. The primary aim of this study was not to compare SLE patients with healthy controls, but to characterize ocular surface changes and their relationship with vision-related quality of life specifically within the SLE population. Nevertheless, I agree that the lack of a control group limits the generalizability of the findings, and this has been clearly addressed in the Discussion as part of the study’s limitations.
Comments 1:
Demographics is repeated three times.
Response 1:
I thank the Reviewer for this observation. I have revised the manuscript to avoid redundancy by removing repeated demographic data from the Methods section and presenting full demographic characteristics only in the Results section.
Page 7, lines 256-258
Comments 2:
Mean age in methods and results are different (46 vs. 51).
Responses 2:
I thank the Reviewer for identifying this inconsistency. The correct mean age is 46 years (SD = 10.75, range 24–65 years), and I have corrected the discrepancy so that the data are now consistent between Methods and Results.
Page 7, lines 257-258
Comments 3:
Inclusion and exclusion criteria are not provided.
Response 3:
I agree with the Reviewer. I have now added explicit inclusion and exclusion criteria to the Methods section to clarify patient selection. These criteria specify that included patients had a confirmed diagnosis of SLE, were aged ≥18 years, and underwent complete ophthalmological assessment. Exclusion criteria included history of ocular surgery, concurrent systemic or autoimmune diseases other than SLE, contact lens use, and incomplete examination data.
Page 3, lines 90-96
Comments 4:
Contact lens wearing was not accounted neither excluded.
Response 4:
I appreciate this remark. I have now specified in the Methods that patients wearing contact lenses were excluded from the study to avoid potential confounding effects.
Page 3, line 95
Comments 5:
The study is focused on the high risk group (premenopausal or menopausal females) where dry eye is to be expected.
Response 5:
I agree with this observation. I have emphasized in the Discussion that the cohort consisted predominantly of premenopausal and menopausal females, which represents a group at increased baseline risk for dry eye disease.
Page 14, lines 469-472
Comments 6:
Female predominance does not allow extrapolation these data on general population.
Response 6:
I agree with this limitation. I have underlined in the Discussion that the female predominance in the study cohort limits the extrapolation of the results to the general SLE population.
Page 13, lines 469-472
Reviewer 3 Report
Comments and Suggestions for Authors
please find attached

Author Response
Comments 1:
The discussion and introduction of results are mixed together. The introduction is a bit of a mishmash. Introduction: Please concentrate on the SLE caused ocular surface complications.
Response1 :
I thank the Reviewer for this remark. I have revised the Introduction by simplifying the description of systemic lupus erythematosus and focusing specifically on ocular surface complications, particularly dry eye disease. I have also clarified the rationale for the study. This restructuring ensures that the Introduction is now more concise and better aligned with the study’s scope.
Page 2, lines 46-53 and 77-79
Comments 2: The analysis used data from both eyes of one patient: show what statistical test was used to ensure that the analysis was not distorted. It is problematic if both eyes of the same patient were used in the statistical analysis. Are there any significant differences between right and left eyes in the same individual for the examined tests, is there a variance test for it?
Response 2: I thank the Reviewer for raising this important methodological point. In accordance with commonly accepted practice in ophthalmological research, when both eyes were examined, the mean value of the parameters obtained from the right and left eye was calculated for each patient. This approach minimized the potential influence of interocular asymmetry and ensured that each patient contributed equally to the analysis, thereby avoiding overrepresentation of individual cases. I have clarified this procedure at the end of the Statistical Analysis section.
Page 7, lines 248-252
Comments 3: What tests would be truly relevant in the early detection of DED?
Respone 3: I appreciate this valuable suggestion. In response, I have expanded the Discussion to address the potential role of early diagnostic tools, such as tear film osmolarity testing, non-invasive TBUT, and meibomian gland imaging. These methods may improve sensitivity in detecting early ocular surface changes and could be incorporated into future research and clinical practice in SLE patients.
Page 13-14, lines 449-468
Comments 4: Statistical analysis: Define exactly which groups were compared and which parameters were examined using which tests.
Response 4: I thank the Reviewer for this comment. The details of the statistical methodology, including the use of parametric and non-parametric tests for continuous variables, chi-squared and Fisher’s exact tests for categorical variables, as well as Pearson’s and Spearman’s coefficients for correlations, were already specified in the Statistical Analysis section of the original manuscript. These tests were consistently applied to compare the DED and non-DED groups and to examine the associations between subjective (OSDI) and objective clinical parameters.
Page 7, lines 228-246
Comments 5: Results: What is the difference between lens opacification and cataract?
Response 5: I thank the Reviewer for this observation. In our study, “lens opacification” was used to describe early, partial lens changes observed during slit-lamp examination, without clinically significant visual impairment, while “cataract” was reserved for advanced, clinically significant opacities associated with reduced visual acuity. For this reason, the distinction was retained in the tables to reflect different stages of lens pathology, although the terminology was not further expanded in the text to avoid overcomplicating the Results section.
Comments 6: Has retinitis/vasculitis ever occurred?
Response 6: I thank the Reviewer for this question. In the examined group, no cases of retinitis or retinal vasculitis were observed. This has been clarified in the Results section.
page 7, lines 269-271
Comments 7: Is there an association between the severity of the described posterior segment abnormalities and DED?
Response 7: I acknowledge this point. No statistically significant association was found between posterior segment abnormalities and the presence of DED in our cohort. This has been clarified in the Results section.
Page 7, lines 269-271
Comments 8: What was the disease duration in the groups? Is there a difference between the two groups in the disease duration?
Response 8: I thank the Reviewer for this important remark. Complete data on disease duration were not available for all patients, which limited the possibility of a full statistical analysis. However, it should be noted that no significant differences in age were observed between the DED and non-DED groups. Since age is one of the strongest risk factors for dry eye disease, this finding suggests that disease duration alone may not be the decisive factor in the occurrence of ocular surface alterations in SLE. I have clarified this point in the Discussion and acknowledged the lack of complete data on disease duration as a limitation.
Page 14, lines 472-478
Comments 9: Lines 302–310, 336–345 and 349–351: These findings are subject to discussion.
Response 9: I thank the Reviewer for this remark. I respectfully decided to retain these statements in the Results section, as they present the mean values and correlations together with their immediate clinical implications. In my view, this structure improves the readability of the results and provides the reader with a direct understanding of the practical relevance of the findings. The broader interpretation and comparison with existing literature are provided in the Discussion, whereas the Results section is limited to the observed outcomes and their direct consequences.
Comments 10: Discussion: Lines 356–365 These statements are subject to Introduction.
Resposne 10: I thank the Reviewer for this remark. In the revised version of the manuscript, the background information on the definition of dry eye disease has been updated to reflect the most recent TFOS DEWS III consensus and placed appropriately at the beginning of the Discussion, immediately after the summary of the study findings. At the same time, the Introduction was simplified and refocused on ocular surface complications in SLE. This restructuring ensures that the Discussion is centered on the interpretation of the present results, while the Introduction provides only the necessary context.
Page 12, lines 368-383
Comments 11: Please restructure the Discussion: It is nothing new to say that DED impairs quality of life. A much more exciting question is what tests we should use to detect DED as early as possible and start treating it appropriately.
Response 11:
I thank the Reviewer for this valuable recommendation. I have restructured the Discussion so that it begins with a summary of the study’s findings and then addresses their broader implications. In response to the Reviewer’s suggestion, I have expanded the Discussion to include a new paragraph emphasizing the potential role of early diagnostic tests such as non-invasive TBUT, tear film osmolarity testing, and meibography of the meibomian glands. These methods may enhance early detection of DED in SLE patients and support timely treatment interventions.
Page 13-14, lines 461-468
Round 2
Reviewer 2 Report
Comments and Suggestions for Authors
All my comments were addressed. Thnal you!
Author Response
I sincerely thank the Reviewer for the positive feedback and for the valuable comments that helped me improve the manuscript.
Reviewer 3 Report
Comments and Suggestions for Authors
Thank you for consideration of my suggestions
Author Response

(The authors gave the same response as above.)
